# Prevalence and Clinicopathological Characteristics of Moderate and High-Penetrance Genes in Non-BRCA1/2 Breast Cancer High-Risk Spanish Families

**DOI:** 10.3390/jpm11060548

**Published:** 2021-06-12

**Authors:** Maria Fonfria, Inmaculada de Juan Jiménez, Isabel Tena, Isabel Chirivella, Paula Richart-Aznar, Angel Segura, Ana Beatriz Sánchez-Heras, Eduardo Martinez-Dueñas

**Affiliations:** 1Cancer Genetic Counseling Unit, Medical Oncology Department, Castellon Provincial Hospital, 12002 Castellon, Spain; mfonfria@uji.es (M.F.); tenagarcia.Isabel@gmail.com (I.T.); eduardo.martinez@hospitalprovincial.es (E.M.-D.); 2Molecular Biology Unit, Service of Clinical Analysis, La Fe University Hospital, 46026 Valencia, Spain; 3Medical Oncology Department, INCLIVA Biomedical Research Institute, University of Valencia, 46001 Valencia, Spain; chirivella_isa@gva.es; 4Cancer Genetic Counseling Unit, Medical Oncology Department, La Fe University Hospital, 46026 Valencia, Spain; paularichart@gmail.com (P.R.-A.); segura_ang@gva.es (A.S.); 5Cancer Genetic Counseling Unit, Medical Oncology Department, Elche University Hospital, 03203 Elche, Spain; sanchez_ana@gva.es

**Keywords:** hereditary breast and ovarian cancer, germline testing, BRCA1 or BRCA2 negative, moderate penetrance genes, next-generation sequencing

## Abstract

(1) Background: Over the last decade, genetic counseling clinics have moved from single-gene sequencing to multigene panel sequencing. Multiple genes related to a moderate risk of breast cancer (BC) have emerged, although many questions remain regarding the risks and clinical features associated with these genes. (2) Methods: Ninety-six BC index cases (ICs) with high-risk features for hereditary breast and ovarian cancer (HBOC) and with a previous uninformative result for *BRCA1/2* were tested with a panel of 41 genes associated with BC risk. The frequency of pathogenic variants (PVs) was related to the clinical characteristics of BC. (3) Results: We detected a PV rate of 13.5% (excluding two cases each of *BRCA1* and *MUTYH)*. Among the 95 assessed cases, 17 PVs were identified in 16 ICs, as follows: *BRCA1* (*n* = 2), *CHEK2* (*n* = 3), *ATM* (*n* = 5), *MUTYH* (*n* = 2), *TP53* (*n* = 2), *BRIP1* (*n* = 1), *CASP8* (*n* = 1), and *MSH2* (*n* = 1). We also identified a novel loss-of-function variant in *CASP8,* a candidate gene for increased BC risk. There was no evidence that the clinical characteristics of BC might be related to a higher chance of identifying a PV. (4) Conclusions: In our cohort, which was enriched with families with a high number of BC cases, a high proportion of mutations in *ATM* and *CHEK2* were identified. The clinical characteristics of BC associated with moderate-risk genes were different from those related to *BRCA1/2* genes.

## 1. Introduction

Breast cancer (BC) is the most frequent cancer type in women and one of the main causes of female mortality. Each year, 2.08 million new cases are diagnosed and there are approximately 626,679 deaths secondary to BC worldwide [1]. In Spain, there are 25,215 new cases of BC per year, making BC the leading cause of cancer-related mortality in women (15.5% of deaths caused by cancer). It is estimated that 5–10% of BC cases have hereditary causes, but among individuals seeking clinical evaluation for hereditary BC, pathogenic variants (PVs) or suspected PVs were only found in *BRCA1* and *BRCA2* (*BRCA1/2*) in 9–29% of cases [2]. In addition, rare germline variants in known high-risk genes such as *TP53, CDH1, STK11,* and *PTEN* predispose to well-defined hereditary cancer syndromes in which BC also develops [3].

In the last decade, advances in sequencing methods have revealed a marked heterogeneity in the loci related to hereditary breast and ovarian cancer (HBOC), with more than 25 genes emerging, most of which are involved in maintenance and repair genome pathways connected with *BRCA1/2* [4]. These moderate-risk genes confer a two to five-fold higher risk of BC, have a variable penetrance, and their expression can be modified by many factors such as other genes, family history, and environmental influences [5]. Several publications in recent years have found that the most frequently altered genes are *CHEK2*, *ATM,* and *PALB2*. While the role of *BRCA1/2* in HBOC is well established, the role of new emerging factors in the development of HBOC has not yet been fully defined, nor has the role they can play in the tumor phenotype. Recent studies have reported that in families with suspected HBOC, almost 50% of the PVs detected were identified in genes other than *BRCA1/2* [6]. Most notable was that no gene was detected in 64–86.5% of women who were clinically evaluated for hereditary BC [7]. In addition, it has been suggested that tumors secondary to mutations in gene pathways similar to *BRCA1/2* could present an increased sensitivity to PARP inhibitors [4].

The identification of new genetic variants using gene panels offers new possibilities for patients and families with suspected HBOC, but also offers new challenges for clinical management and genetic counseling. Multiple commercial panels have been developed, with each one covering a variable number of high- or moderate-risk genes with single nucleotide polymorphisms (SNPs), alongside candidate genes whose risk is not yet clearly defined in current guidelines [5]. Considering these factors, several publications have warned about the lack of regulation of these panels and the multiple clinical and ethical considerations associated with their use [5,8,9]. To establish the clinical utility of these panels, we must first understand the prevalence, penetrance, and phenotypes of BC predisposing mutations other than *BRCA1/2* in order to make the best use of these data in genetic clinical services. The use of multi-gene panels in research protocols allows researchers to explore the epidemiology and causal role of these genes and to assess which ones should be offered in the context of genetic counseling.

In this current work, we analyzed a population with a high-risk of HBOC and without *BRCA1/2* mutations to determine the prevalence and penetration of mutations in other genes predisposing these patients to BC. We also analyzed the tumor phenotype present in mutations other than *BRCA1/2* in order to obtain prognostic and clinical–pathological information.

## 2. Materials and Methods

### 2.1. Patient Selection

The study participants were retrospectively selected from a population of BC patients who met the criteria for high-risk HBOC provided by the *Comunidad Valenciana* Cancer Plan in Spain. These patients had been previously assessed in genetic cancer clinics at different hospitals in the *Comunidad Valenciana* and had uninformative test results for their *BRCA1/2* status. The high-risk criteria for BC used to refer patients to genetic clinics between 2005 and 2016 were used for patient selection, although we prioritized families with a larger number of BC cases. These criteria were (1) families with three or more cases of breast or ovarian cancer (OC) among their first or second-degree relatives; (2) two first-degree relatives with BC diagnosed before the age of 50 years; (3) two first-degree relatives with BC, one of them bilateral BC and the other diagnosed before the age of 40 years, and (4) two first-degree relatives with one case of BC and one case of OC. We excluded families with just one case of BC at a young age.

The demographic, clinical, and familial histories were collected from genealogical trees and medical records. Informed consent was obtained from all subjects involved in this study. Positive results for any variants with clinical implications were reported to the patient. However, variants with unknown significance (VUS) and alterations in genes with no established clinical management were not reported to patients. This study was approved by the Ethical Review Board at the Provincial Hospital of Castellon.

### 2.2. DNA Extraction

A total of 96 BC index patients with a high risk of BC were selected retrospectively. We extracted DNA from peripheral blood lymphocytes sample for their *BRCA1/2* study and the excess was stored in the Health Care Biobank. The study was performed in genomic DNA extracted from 500 μL of whole blood using the MagNA Pure LC DNA Isolation Kit, large volume (Roche, Mannheim, Germany) automated in the MagNA Pure LC System (Roche), according to the manufacturer’s protocol.

### 2.3. Next Generation Sequencing

We included 11,820 XT target capture probes (Agilent, Santa Clara, CA, USA) on an array (size = 251,139 kbp) for the target enrichment of the entire gene coding regions and all the flanking non-coding regions for the following selected genes: *ATM, BARD1, BLM, BRCA1, BRCA2, BRIP1, CASP8, CDH1, CDK4, CDKN2A, CHEK2, FANCA, FANCC, FANCD2, FANCE, FANCF, FANCG, FANCM, FAM175A, KRAS, MAP3K1, MEN1, MLH1, MSH2, MSH6, MRE11A, MUTYH, NBN, NF1, PALB2, PTCH1, PTEN, RAD50, RAD51B, RAD51C, RAD51D, REQCL, STK11, TGFB1, TP53,* and *XRCC2*. The capture was performed automatically using a Bravo Robot according to the manufacturer’s instructions. 

The enriched libraries were clonally amplified on a solid substrate for next-generation sequencing (NGS) using Illumina V3 chemistry on a MiSeq System (Illumina, San Diego, CA, USA) to an average coverage of 300× with 2 × 150 paired-reads and a minimum coverage of 25× (with a variant allele frequency cutoff of 20.0%) for each targeted position. Reads were aligned to the hg19/GRCh37 reference human genome build using Alissa Interpret (Agilent technologies, Santa Clara, CA, USA). A minimum quality threshold of Q20, which translates into a sequencing accuracy of >90% for all called bases, was applied. All the clinically relevant variants detected were confirmed by Sanger sequencing. 

### 2.4. Variant Classifications

We consulted databases including VarSome, ClinVar, and UMD to classify the variants according to a five-tier system (deleterious: class 5; likely deleterious: class 4; VUS: class 3; likely benign: class 2; and benign: class 1) following recommendations proposed by the American College of Medical Genetics and Genomics (ACMG). We used the nomenclature approved by the HGVS.

### 2.5. Statistical Analysis

The prevalence of the mutations detected and patient characteristics were reported with descriptive statistics. The demographic, clinical, and pathological characteristics were compared using χ^2^ tests for categorical variables and Student’s *t*-tests or analysis of variance for continuous variables. The odds ratio (OR) was compared between the different groups for each clinical factor using Fisher exact tests. Quantitative variables without a normal distribution were tested with the non-parametric alternative (Mann–Whitney U or Wilcoxon tests). *p*-values < 0.05 were considered significant in every case.

## 3. Results

### 3.1. Characteristics of the Study Population

We selected 96 BC index patients (92 women and 4 men) from Spanish families with a high risk of BC from four departmental hospitals. As shown in Figure 1, our study population was enriched with patients with a high number of BC cases in their family history (median number = 3, range = 1–10). In addition, 16% of the families reported cases of OC, 16% had prostate cancer cases, 6% had pancreatic cancer cases, and 8% had colon cancer cases. 

The median age at BC diagnosis was 46.0 years (range 23–73), with most IC (75%) having been diagnosed before 50 years of age, 8% of which were patients aged <30 years. In addition, 18% of these patients had had bilateral BC; 75% of the tumors were hormone receptor-positive and 16% had a *HER2* amplification. Information regarding HER2 was unavailable in 18% of cases, mainly when patients had been diagnosed before HER2 was routinely determined (before the year 2000). The patient characteristics are summarized in Table 1.

### 3.2. Next-Generation Sequencing Quality

Among the 96 BC ICs selected for analysis, 95 passed the sequencing quality filters for further data analysis. The average and median read depth in analyzable target regions was 65x (range = 50–115); the mean percentage of analyzable target regions covered by at least 50 reads was 98.8%, with 95.58% being covered at least by 100 reads (Appendix A). All the variants were detected in a deep >25× (range = 30–100×) and PV were confirmed by Sanger sequencing. 

### 3.3. Frequency of Deleterious Mutations

Among the 95 cases studied, 17 PVs were identified in 16 ICs as follows: *BRCA1* (*n* = 2), *CHEK2* (*n* = 3, one frameshift, and two stop codon), *ATM* (*n* = 5, two splicing, one frameshift, and two stop codon), *MUTYH* (*n* = 2), *TP53* (*n* = 2, two missense), *BRIP1* (*n* = 1), *CASP8* (*n* = 1) and *MSH2* (*n* = 1). We identified one woman with two PVs, one each in the *ATM* and *CHEK2* genes. We also found 26 VUS in 22 ICs (22.9%). VUS are listed in Appendix A. Indeed, five of the ICs harbored two variants (one patient with two PVs, one patient with a PV and a VUS and three patients with two VUS). 

In addition, we found two PVs in the *BRCA1* gene. In one patient, the c.5324T > G (*p*.Met1775Arg) variant had been previously classified as a VUS when she had first been evaluated at the genetic cancer clinic, but this VUS has since been reclassified as a PV. The PV detected in the other patient was *BRCA1* NM_007294.3 c.5152 + 5G > A and had remained undetected in her first study. According with these new classifications or detections, we contacted these two women and offered them updated recommendations and familial screening. We also detected two monoallelic pathogenic mutations in *MUTYH*.

Frequencies of PVs and VUS are summarized in Figure 2. Excluding these two *BRCA1* mutations and the two monoallelic *MUTYH* mutations, our PV rate in non-*BRCA1/2* genes was 13.5%. The most prevalent variants detected were in *ATM* (5.2%) and *CHEK2* (3.1%), with mutations in these two genes accounting for 47% of the all the PVs detected. The remaining PVs were in *TP53* (2.08%), *BRIP1* (1.04%), *MSH2* (1.04%), and *CASP8* (1.04%), as described in Table 2.

Furthermore, we found two unrelated index BC cases with the same PV (NM_00051.4:c.2921 + 1G > A) in *ATM* (rs587781558). One IC harbored the *CHEK2* variant (NM_007194.4:c.1036C > T) (rs201206424), a missense variant classified as a VUS by ClinVar and as likely pathogenic by VarSome. This sequence replaces arginine with cysteine at codon 346 of the CHEK2 protein (*p*.Arg346Cys) and has been reported to affect CHEK2 protein function [10]. In addition it has been observed in individuals affected with BC [11]. However, the available evidence is currently insufficient to determine the role of this variant in disease and therefore, it was classified as a VUS (or C:3).

### 3.4. Clinical Characteristics in Cases with Pathogenic Variants

In terms of different age ranges, PVs were present in 13.6% of individuals aged under 45 years (*n* = 44) and in 11.8% of those aged over 45 years (*n* = 50; χ^2^ = 0.075, *p* = 0.784 after excluding cases with a PV in *BRCA1/2* or *MUTYH*). The median age was 45 years (range 28–66) among patients with a PV and 46 years (range 23–73) in those with no mutation, with no significant differences observed between these groups. In addition, the number of BC cases in the family or the presence of bilateral BC was not a predictor for the detection of a PV. Moreover, a correlation analysis using Fisher tests found no link between the clinical factors we evaluated and an increased risk of detecting a PV (Table 3).

After excluding patients with PVs in *BRCA1* and *MUTYH*, we identified a pathogenic mutation in 9 of the 67 patients who were hormone receptor-positive (13.4%). Information regarding HER2 status was available for a subgroup of 80 patients; we detected a mutation in two (12.5%) of the sixteen patients with a *HER2* amplification. We were only able to confirm seven cases with triple negative BC, among which a mutation in *BRCA1* and no pathogenic mutation in non-*BRCA* genes were detected (Table 4). All the cases of BC with *CHEK2* mutations were hormone receptor-positive and one harbored an amplified *HER2*, but we did not identify any cases with the founder mutation c.1100delC. Of note, the families of these ICs had experienced a high number of BC cases (between 3 and 10). In addition, some authors have suggested an association with colon cancer for *CHEK2* mutations [12]. One of our ICs had metachronous breast and colon cancer at a young age. No other cases of colon cancer were described in the other families of patients with the *CHEK2* mutation.

Furthermore, ICs with mutations in *ATM* presented BC with hormonal receptor expression and without HER2 amplification. The average number of BC cases in *ATM* families was lower than for *CHEK2* (between two to three cases of BC per family) and one of these families had two cases of OC. Recent data have described an association between *ATM* and OC [13].

Two PVs were identified in *TP53*; the two ICs were diagnoses of BC at a notably young age (28 and 32 years) and one of them had a *HER2* amplification. However, the information regarding HER2 was unavailable in the other case because the initial BC diagnosis had been made before HER2 determination was routine. We identified one pathogenic mutation in *BRIP1* in a family without any reported cases of OC. Finally, we also identified a PV in *MSH2* in a case of a family with two or more cases of BC and one case of colon cancer, without meeting the Bethesda or Amsterdam criteria; in addition, this IC harbored a VUS in *RAD50*.

## 4. Discussion

All the PVs detected in our series have been previously described as disease-causing agents (BC families, Li-Fraumeni syndrome, Lynch syndrome, or ataxia-telangiectasia) [11,14,15,16,17,18,19,20,21,22,23,24], except for the *BRIP1* NM_032043.2:c.508-2A > T (rs876659797) and *CASP8* NM_01228.4: c.331delG (rs776712453). The *BRIP1* variant has not been reported previously in the literature in individuals with BRIP1-related disease [25]. The *CASP8* variant is a novel variant that has been classified as likely pathogenic in VarSome but has not yet been described in ClinVar; it is a rare variant with an allele frequency of 0.000024 in GnomAD (exomes). The *MUTYH* monoallelic mutation rate of 2% is similar to the rate that would be expected in the general population [26].

In our series, the PV detection rate was higher than that reported elsewhere. In a published paper that includes patients with BC without hereditary criteria, mutation rates of 4.1% have been reported in non-*BRCA1/2* genes. [27]. Studies in families with a high risk of BC have reported a mutation rate of between 4% and 6.2% in non-*BRCA1/2* genes [8,28,29]. Although the series in American populations of high-risk families present data for a remarkably high number of patients, relatively less information is available for the European population, and especially, the Spanish population. Recently, Bonache et al. published a study in 300 high-risk Spanish families in which the non-*BRCA1/2* gene mutation rate was 8% [30]. Although *ATM* and *CHEK2* were the most frequent PVs detected in the HBOC population, the high frequency detected in our series for these genes was striking. In the large series by Couch et al. of 58,798 women with BC that were referred to hereditary cancer clinics, a mutation rate of 1.73% for *CHEK2*, 1.06% for *ATM,* and 0.87% for *PALB2* were reported [28]. In a German cohort of 5589 BC index patients, with hereditary high-risk criteria, Hauke et al. found a mutation prevalence of 2.5% for *CHEK2*, 1.5% for *ATM,* and 1.2% for *PALB2* [31]. The elevated percentage of PVs in ATM and CHEK2 in our study may be because families had a high burden of BC.

The five pathogenic mutations we detected in *ATM* (two splicing, one frameshift, and two stop codon) caused a loss of function (LoF) of the protein. Of the four PVs detected in *CHEK2*, three caused a LoF (one frameshift and two stop codon) and the other was a missense mutation. A recent study assessed whether there is a different risk of BC according to the type of mutation detected and found that for ATM, this risk is higher in variants with LoF than in deleterious missense variants. However, they found no differences for the types of variants in *CHEK2* and *PALB2* genes. [32] Another recent paper also reported that rare missense variants in *CHEK2* were associated with an increased risk of BC, and that this association was independent of the locus [33]. We found a rare missense variant in *CHEK2* with discrepancies in the classification guidelines, but with some evidence of pathogenicity reported in the literature and predictive models [10,11].

Of note, we detected no PVs and only one VUS in *PALB2*, probably because of the small size of our series. Indeed, this gene has been repeatedly identified as mutated in women with BC and family risk factors. In fact, Thompson et al. found that half the risk resulting from mutations in genes other than *BRCA1/2* was caused by mutations in *PALB2* [8]. Regarding *ATM* and *CHEK2*, deleterious mutations in these genes were more frequent in estrogen receptor (ER)-positive tumors, while *CHEK2* mutations were also frequently found in HER2 positive tumors [13,31,33]. Consistent with these data, all the BC cases in our series with mutations in one of these two genes were ER-positive.

Both missense *TP53* variants identified (c.743G > A, *p*.Arg248Gln, and c.638G > A, *p*.Arg213Gln) are listed in the IARC *TP53* database [34] and have been reported as dominant-negative PV (in which the mutated p53 protein interferes with the function of the wild type p53 protein) [35]. Some studies have reported that individuals with dominant-negative PV appeared to have more clinically severe phenotypes than individuals with other *TP53* PV did [36,37]. None of the two families in our study fulfilled the classic criteria for Li-Fraumeni syndrome, although they meet the Chompret criteria [38,39]. In addition, the two ICs carrying a *TP53* mutation were diagnosed with BC at a very young age and one of them harbored a *HER2* amplification. Mutations in *TP53* are associated with early-onset BC and over two-thirds of BCs in women with Li-Fraumeni syndrome have some degree of *HER2* amplification [40]. Some authors have proposed that cases of BC in very young women (before 31 years) should be tested for *TP53* genes because, in many families, the mutations in this gene would not have been detected via a family history alone [36]. In these cases, the identification of a pathogenic *TP53* mutation allows physicians to change the management in young unaffected females carrying this mutation; for example, by suggesting they undergo a risk-reducing bilateral mastectomy [41]. 

One of the ICs we report here presented a pathogenic mutation in *MSH2* in a family with two cases of BC, one case of colon cancer diagnosed at 68 years, and one of gastric cancer diagnosed at 55 years. Although multiple reports have suggested a moderate risk of BC in cases of mismatch repair gene mutations, especially for *MSH6* [13], other authors did not observe this relationship [28]. Furthermore, current National Comprehensive Cancer Network guidelines do not advise following patients with Lynch syndrome for BC, but rather recommend management based solely on the family history of BC [3]. However, the relationship between Lynch syndrome and OC is more robust, although its relative risk has not yet been clearly established; Lu et al. observed that mismatch repair genes confer a moderate risk for OC, with an OR 4.16 [13].

Although *BRIP1* is related with a moderate risk for OC and its relationship with BC is not well established, we found a PV in a family with high-risk features for BC and without any case of identified OC. A similar finding was also reported in another Spanish cohort [30].

One BC IC harbored a novel LoF variant in *CASP8*. This frameshift variant has not been described previously in ClinVar and was predicted to encode for a non-functional protein in VarSome. *CASP8* regulates apoptosis and common variants of this gene (SNPs) have been identified as low penetrance BC genes through genome-wide analysis in several trials [42,43,44,45]. Although little is known about germinal LoF variants in *CASP8* in relation to BC risk, somatic mutations in this gene have been identified as drivers of BC development [46]. Recent studies have proposed that rare LoF variants from low penetrance genes in BC, including *CASP8*, could be associated with a higher risk of BC than SNPs [47,48]. However, because insufficient data are available, this result will not be used in clinics and a segregation analysis was instead proposed for this family.

Consistently with already published extensive series, which do not consider *RAD50, NBN, MLH1, PMS2,* or *MRE11* relevant factors in BC risk [8,28], we did not identify any PVs in any of these genes. However, *RAD51C* and *RAD51D* were related with a moderately increased risk in two large, recently published studies [33,49]. 

The high percentage of VUS in these moderate-penetrance genes is another recurrent characteristic described in these studies, with reported ranges of 14–36% [27,31,50]. We found VUS in 22.9% of our cohort, which is within the range commonly described. VUS should not be used for clinical decision making, but their high percentage is expected to decrease with further research over time, in an analogous way to those for the *BRCA1/2* genes.

One of the advantages of multi-gene panel sequencing is that it allows patients carrying more than one mutation to be detected where these individuals would have previously remained undetected by the sequential study of mutations. Therefore, these tests allow physicians to provide patients with more complete genetic advice [29]. For example, in our work, we detected one patient with two PV, one each in *ATM* and *CHEK2*. 

The introduction of multigene panels has led to the frequent detection of genes associated with specific cancer syndromes in individuals that do not meet the inclusion criteria usually established by studies [29]. For example, in our series, we detected one IC with a *MSH2* mutation whose family did not meet the Amsterdam II or Bethesda criteria for Lynch syndrome. Nonetheless, although this may help detect syndromes that were not detected in single-gene sequencing, many authors have warned that the presentation of cases detected by multigene panel testing may be different from that of cases detected by classic syndromic presentations. For example, Rana et al. found that the average age of presentation of BC in patients with mutations in *TP53* detected by a multigene panel was greater than the mean age of those detected by analysis of a single gene [51]. In fact, some authors have warned that in some cases, the detection of these PVs may be an incidental finding not causally related to the family risk of BC, and so these findings should be interpreted with caution, both towards patients and their families. Indeed, Thompson et al. found a similar rate of mutations between a BC population enriched for hereditary cancer features and a cancer-free control population [8].

It is also important to remember that criteria that predict the probability of mutation in non-*BRCA1/2* genes are largely unknown and may differ from those that predict mutation probabilities in *BRCA1*/2 [31]. Current testing guidelines do not adequately account for the full range of clinical presentations described to date as associated with BC, and carriers of clinically actionable variants in genes other than *BRCA1/2* are likely to still fall outside these guidelines [52]. Recent studies have found no relationship between a younger age at the time of BC diagnosis and a higher prevalence of PVs in moderate-penetrance genes [31,50,53]. Accordingly, we observed no difference in median age between patients with or without PV. Another remaining question is whether a higher family burden of BC cases implies a greater probability of detecting PVs. Our population mainly comprised patients selected because they had a strong family history of BC, which could perhaps partially justify the high rate of mutation detected. However, we did not observe that the detection of a greater number of cases of BC in the family implied an increased probability of detecting PVs.

Considering all these issues together, although multigene panel testing provides large, pragmatic data sets, at the individual level, this information must be interpreted with caution to avoid providing patients with potentially misleading clinical misinformation that could cause harm [8]. The value of multigene panel testing remains controversial, because there is uncertainty regarding the strength of association between mutations in some genes and the development of cancer (clinical validity) and there is a lack of genetic evidence demonstrating improved outcomes for the individuals tested (clinical utility) [9]. Clinicians must possess robust data with proven utility before using genetic testing involving new BC risk genes to provide adequate and proportional risk reduction strategies and to avoid the overtreatment of these families [41]. Regarding intervention measures, no scientific evidence on the role of risk-reducing surgeries is yet available, either at the level of the breast or ovary, and some authors have warned of the risk of causing harm to the patient if measures taken for high-penetrance genes are applied in patients carrying mutations in moderate-penetrance genes [8]. In fact, the review by Tung et al. proposes guidelines for monitoring and evaluating surgeries based on the risk of each gene and the existence of a significant family history [9].

In summary, the cohort considered in our work was small compared to the studies with thousands of patients presented in recent years, with the latter allowing much more robust conclusions to be drawn. However, given the many questions that remain to be answered in the field of moderate-penetrance genes in HBOC and the fact that these large-scale studies included patients that represented populations different from ours, we suggest that collaborative efforts with population studies from different areas could help researchers to better understand this syndrome. Notwithstanding, working on this ‘small patient sample’ allowed us to familiarize ourselves with the technique of NGS and validate it. Thus, we recognized genes that we had never previously studied which are now part of our routine clinical practice assessments. In fact, many of these genes have now been incorporated into the service portfolio of the Genetic Counseling Program forming part of the *Comunidad Valenciana* Cancer Plan. An NGS panel with 40 genes related with hereditary cancer syndromes was developed in our community. From this panel, a subgroup of 13 genes is analyzed in those patients with HBOC syndrome: *ATM, BRCA1, BRCA2, BRIP1, CHEK2, EPCAM, MLH1, MHS2, MSH6, NBN, PALB2, RAD51C,* and *RAD51D. CDH1, NF1, PTEN, STK11* and *TP53* genes are analyzed when the family meets clinical criteria for it. All these genes were included in our previous panel here presented.

## Figures and Tables

**Figure 1 jpm-11-00548-f001:**
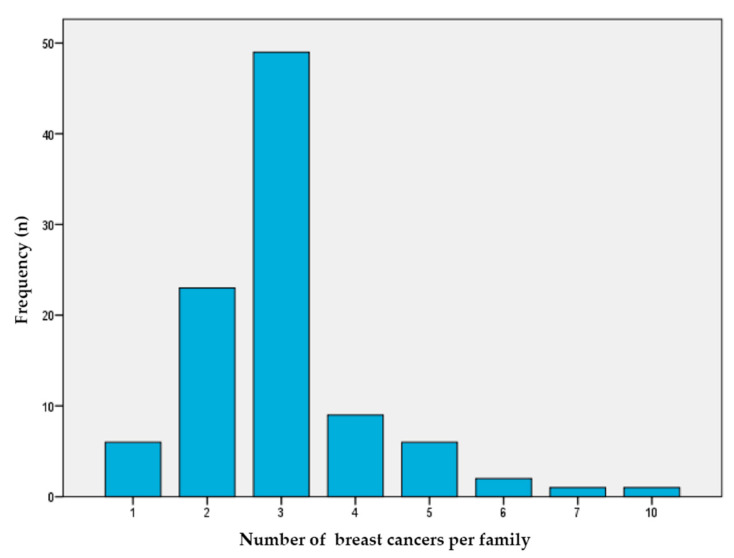
Number of breast cancer cases per family. This figure shows the number of breast cancer per family (*X* axis) that each index case in the study presents (*Y* axis).

**Figure 2 jpm-11-00548-f002:**
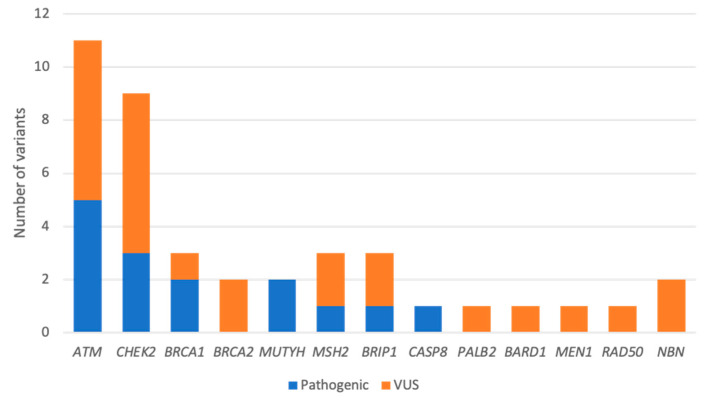
Distribution of germline variants detected by an NGS panel of 41 genes. Variants are classified as pathogenic (Class 4 or 5) or VUS (Class 3).

**Table 1 jpm-11-00548-t001:** Characteristics of study population.

	All	With Mutation ^1^	Without Mutation
Total patients (*n*)	95	14	81
Sex			
Female	91 (95.7%)	13 (92.8%)	78 (96.2%)
Male	4 (4.2%)	1 (7.14%)	3 (3.7%)
Median age at diagnosis (range)	46 (23–73)	45 (28–66)	46 (23–73)
Age at diagnosis			
<30	8 (8.4%)	1 (7.1%)	7 (8.6%)
30–40	22 (23.1%)	4 (28.5%)	18 (22.2%)
41–50	43 (45.2%)	6 (42.8%)	37 (45.6%)
51–60	17 (17.8%)	1 (7.1%)	16 (19.7%)
61–70	5 (5.2%)	1 (7.1%)	4 (4.9%)
>70	1 (1.0%)	0	1 (1.2%)
T stage			
Tis	9 (9.4%)	0	9 (11.1%)
T1	43 (45.2%)	4 (28.1%)	39 (48.1%)
T2	31 (32.6%)	4 (28.1%)	27 (33.3%)
T3	8 (8.4%)	2 (14.2%)	5 (6.1%)
T4	2 (2.1%)	1 (7.1%)	1 (1.2%)
N stage			
N0	56 (58.9%)	4 (28.5%)	52 (64.1%)
N1	30 (31.5%)	5 (35.7%)	25 (30.8%)
N2–N3	8 (8.4%)	3 (21.4%)	5 (6.1%)
Grade			
1	22 (23.1%)	5 (35.7%)	17 (20.9%)
2	41 (43.1%)	6 (42.8%)	35 (43.2%)
3	21 (22.1%)	1 (7.1%)	20 (24.6%)
Unknown	13 (13.6%)	2 (14.2%)	11 (13.5%)
Hormonal receptor status			
Positive	74 (77.8%)	11 (78.5%)	63 (77.7%)
Negative	14 (14.7%)	2 (14.2%)	12 (14.8%)
Unknown	8 (8.4%)	1 (7.1%)	7 (8.6%)
Estrogen receptor status			
Positive	71 (73.6%)	9 (64.2%)	61 (75.3%)
Negative	17 (17.8%)	4 (28.5%)	14 (17.2%)
Unknown	7 (8.4%)	1 (7.1%)	6 (7.4%)
Her2 status			
Positive	16 (16.8%)	2 (14.2%)	14 (17.2%)
Negative	63 (66.3%)	10 (71.4%)	53 (65.4%)
Unknown	18 (18.9%)	2 (14.2%)	16 (19.7%)
Immunophenotype			
Luminal A	35 (36.8%)	6 (42.8%)	29 (35.8%)
Luminal B	22 (23.1%)	3 (21.4%)	19 (23.4%)
Triple negative	7 (7.3%)	1 (7.1%)	6 (7.4%)
Her2 + Hormonal Receptor -	5 (5.2%)	1 (7.1%)	4 (12.3%)
Triple positive	11 (11.5%)	1 (7.1%)	10 (12.3%)
Unknown	17 (17.8%)	2 (14.2%)	15 (18.5%)
Bilateral breast cancer	18 (18.9%)	3 (21.4%)	15 (18.5%)
Family history of cancer			
Breast	95 (100%)	14 (100%)	81 (100%)
Melanoma	2 (2.1%)	0	2 (2.4%)
Ovarian	17 (17.8%)	2 (14.2%)	15 (17.8%)
Prostate	17 (17.8%)	2 (14.2%)	15 (17.8%)
Colorectal	9 (9.4%)	3 (21.4%)	6 (7.4%)
Pancreatic	5 (5.2%)	1 (7.1%)	4 (4.7%)
Others	13 (13.6%)	3 (21.4%)	10 (12.3%)

^1^ Two ICs with monoallelic *MUTYH* mutation have been included in the “no mutation subgroup” because of the recessive inheritance of this mutation.

**Table 2 jpm-11-00548-t002:** Pathogenic variants in the study cohort.

Gene Variants									Personal History			Family History		
Study ID	Gene	Variant Type	Class ^1^	HGVS Coding	HGVS Protein	Transcripts	rsID	2nd Variant	BC Age	Subtype (ER/PR/HER2)	Bilateral BC	No. Cancers	No. BCs	No. OCs
35022	*BRCA1*	Splicing	5	c.5152 + 5G > A	-	NM_007294.3	rs80358165	-	45	−/+/−	No	2	2	0
30983	*BRCA1*	Missense	4	c.5324T > G	p.Met1775Arg	NM_07300.4	rs41293463	-	31	−/−/−	Yes	1	1	0
24247	*ATM*	Stop codon	5	c.2413C > T	p.Arg805Ter	NM_000051.4	rs780619951	-	34	+/+/−	Yes	3	2	0
45434	*ATM*	Splicing	5	c.2921 + 1G > A	-	NM_000051.4	rs587781558	-	42	+/+/−	No	3	3	0
36845	*ATM*	Splicing	5	c.2921 + 1G > A	-	NM_000051.4	rs587781558	-	45	+/+/−	No	4	2	1
26672	*ATM*	Frameshift	5	c.43delC	p.Leu15Terfs	NM_000051.4	rs771887195	-	48	+/+/−	No	2	2	0
47150	*ATM*	Stop Codon	5	c.4507C > T	p.Gln1503Ter	NM_000051.4	rs1131691164	CHEK2 c.1555C > T (p.Arg519Ter)Class 5	48	+/+/−	No	3	2	1
60766	*CHEK2*	Stop Codon	5	c.279G > A	p.Trp93Ter	NM_007194.4	rs587782070	-	66	+/+/−	Yes	10	10	0
36497	*CHEK2*	Frameshift	5	c.591delA	p.Val198Phefs	NM_007194.4	rs587782245	-	37	+/+/+	No	3	2	0
55930	*TP53*	Missense	5	c.743G > A	p.Arg248Gln	NM_000546.6	rs11540652	-	28	−/−/+	No	3	2	0
18358	*TP53*	Missense	5	c.638G > A	p.Arg213Gln	NM_000546.6	rs587778720	-	32	−/+/?	No	6	3	0
61221	*BRIP1*	Splicing	5	c.508−2A > T	-	NM_032043.2	rs876659707	-	62	+/+/−	No	3	1	0
39477	*MSH2*	Missense	4	c.2320A > G	p.Ile774Phe	NM_000251.2	rs775464903	-	50	+/+/−	No	5	2	0
36984	*MUTYH*	Missense	5	c.1187G > A	p.Gly396Asp	NM_001048174.2	rs36053993	-	50	+/+/−	No	3	3	0
43701	*MUTYH*	Missense	5	c.1187G > A	p.Gly396Asp	NM_001048174.2	rs36053993	-	51	+/−/−	No	4	3	0
38266	*CASP8*	Frameshift	4	c.331delG	p.Ala111Leufs*22	NM_01228.4	rs776712453	NBN c.1238A > G (p.Asn413Ser) Class 3	57	+/+/−	No	7	3	0

BC: breast cancer. ER: estrogen receptor. PR: progesterone receptor. OC: ovarian cancer. Databases used: ClinVar: Clinical Variation (https://www.ncbi.nlm.nih.gov/clinvar/; accessed on 11 February 2021); VarSome: the Human Genomics Community (https://varsome.com/; accessed on 11 February 2021). ^1^ Class: variant classification proposed by the American College of Medical Genetics and Genomics (ACMG) (deleterious: class 5; likely deleterious: class 4; VUS: class 3; likely benign: class 2; and benign: class 1).

**Table 3 jpm-11-00548-t003:** Comparison of the odds ratio (OR) of pathogenic variants (PVs) in different groups for each clinical factor using Fisher exact tests.

Clinical Factors	Level	PV Ratio	OR (95% CI)	*p*-Value
Age	≤45	17.78% (8/45)	1.32 (0.38, 4.74)	0.77927
	> 45	14.00% (7/50)		
Bilateral involvement	No	15.79% (12/76)	0.94 (0.21, 5.82)	1
	Yes	16.67% (3/18)		
Luminal	No	15.00% (6/40)	0.90 (0.24, 3.16)	1
	Yes	16.36% (9/55)		
*Her2* positive	No	19.05% (12/63)	1.64 (0.31, 16.78)	0.72285
	Yes	12.50% (2/16)		
Triple negative	No	15.91% (14/88)	1.13 (0.12, 55.90)	1
	Yes	14.29% (1/7)		
Family ovarian cancer	No	16.25% (13/80)	1.26 (0.24, 12.80)	1
	Yes	13.33% (2/15)		
Stage	0	0.00% (0/10)		0.12418
	I	13.79% (4/29)		
	II	13.16% (5/38)		
	III	37.5% (6/16)		
	IV	0.00% (0/1)		

Note: the odds ratios were compared using Fisher exact tests.

**Table 4 jpm-11-00548-t004:** Deleterious mutations by breast cancer (BC) subtype.

Genes	Patients with Luminal BC (*n* = 57)		Patients with *HER2* Positive BC (*n* = 16)		Patients with Triple Negative BC (*n* = 7)	
	No.	% (95% CI)	No.	% (95% CI)	No.	% (95% CI)
Any deleterious mutation	9	9.5 (3.5–15.3)	2	2.1 (0–4.9)	1	1.1 (0.1–3.1)
Genes related to breast cancer						
*BRCA1*	1	1.1 (0.1–3.1)	0		1	1.1 (0.1–3.1)
*ATM*	5	5.3 (0.7–9.7)	0		0	
*CHEK2*	3	3.2 (0.3–6.6)	1	1.1 (0.1–3.1)	0	
*TP53*	1	1.1 (0.1–3.1)	1	1.1 (0.1–3.1)	0	
Genes not clearly related to breast cancer						
*MSH2*	1	1.1 (0.1–3.1)	0		0	
*BRIP1*	1	1.1 (0.1–3.1)				
Candidate genes			0		0	
*CASP8*	1	1.1 (0.1–3.1)	0		0	

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
