# Peer review of "Prevalence and Clinicopathological Characteristics of Moderate and High-Penetrance Genes in Non-BRCA1/2 Breast Cancer High-Risk Spanish Families"

_jpm, 2021, doi:10.3390/jpm11060548_

Round 1

Reviewer 1 Report

It is commendable that the authors have undertaken this type of clinical study and have reported results back to patients. However from a scientific perspective there is a lack of novelty, the sample size is small and the results are less likely to be of interest at an international level.

As the authors point out,  Bonache et al. published a larger study in 300 high-risk Spanish families. The study would have benefited greatly if the tumours had been sequenced in addition to (I’m assuming) germline DNA.

 There are several inconsistencies in the manuscript.

Section 2.2. No details about DNA extraction are provided or the tissue from which DNA was extracted. Tumour, blood, adjacent normal?
The manuscript describes results form female patients in the text but Table 1 includes 4 male patients
Table 1 “Median age at diagnosis” - median or mean?
 The authors compare their results to Couch et al. and Hauke et al (line 298-302) - are the cohorts similar, i.e. are the selection criteria similar? Are the methodologies similar?

Author Response

First of all, I would like to appreciate the time and attention to our manuscript.

Point 1. It is commendable that the authors have undertaken this type of clinical study and have reported results back to patients. However from a scientific perspective there is a lack of novelty, the sample size is small and the results are less likely to be of interest at an international level

Response. Effectively, as you correctly point out, the sample size is small. It is a pilot study that has been used to implement the NGS in our health area. We expect to collect the results of the NGS program implemented in a larger population in the future. Regarding international relevance, we tried to confirm that in our environment there was the mutational profile described in other populations, since there was little information published in Spanish population.

Point 2. As the authors point out, Bonache et al. published a larger study in 300 high-risk Spanish families. The study would have benefited greatly if the tumours had been sequenced in addition (I'm assuming) germline DNA.

Response. We find your contribution of sequencing tumors very interesting. However, at the time of this study, we did not have enough funding for this issue. We collect your idea to propose a second study phase, as published by Wu et al. in the article "Integrating Germline and Somatic Mutation Information for the Discovery of Biomarkers in Triple-Negative Breast Cancer. Int J Environ Res Public Health. 20019 Mar 23; 16(6):1055."

Point 3. There are several inconsistencies in the manuscript. Section 2.2. No details about DNA extraction are provided or the tissue from which DNA was extracted. Tumour, blood, adjacent normal?

Response. Certainly, we have omitted the details about the specimen and method of DNA extraction. We have considered your suggestion and proceed to include these details (DNA extracted from blood) in the revised manuscript (Section 2.2) and generate a new section 2.3 with the NGS technology.

Point 4. The manuscript describes results from female patients in the text, but table 1 includes 4 male patients.

Response. Indeed, we have 4 families in which the index case was a male with breast cancer who meets criteria for hereditary breast and ovarian cancer syndrome and this is not described in the manuscript. We proceed to correct it in the revised manuscript (line section).

Point 5. Table 1. "Median age at diagnosis"- median or mean?

Response. As you point out, there is an inconsistency. In the table 1 we refer to the median, but by mistake we added the mean in the cases of mutation and no mutation. We proceed to correct the error and add the range, as well as describe the median in the manuscript instead of the mean in order to avoid more confusion (section 1.4 Clinical characteristics of patient with pathogenic variation).

Point 6. The authors compare their results to Coach et al. and Hauke et al. (line 298-302), are the cohorts similar, i.e. are the selection criteria similar? Are the methodologies similar?

Response. In both studies patients were referred to hereditary cancer clinics because their personal and family history of breast cancer. In the Hauke et al. study, index cases meet criteria for hereditary breast and ovarian cancer syndrome and they have a previous uninformative BRCA1/2 testing, in a similar way that in our study. The methodology employed in both studies for established the prevalence of germline mutations in moderate genes was NGS, like us. In the Coach et al. study, they present a nationwide sample of 65057 breast cancer patients referred for hereditary cancer genetic testing, but they don't specify if hereditary criteria are fulfilled.  We proceed to clarify these points in the manuscript.

*Additional comment: Following your recommendations, in order to clarify data presentation, in the revised manuscript we have changed "Table 2. Germline PV identified" by "Figure 2. Distribution of variants detected by Next Generation Sequencing panel of 41 genes"

Reviewer 2 Report

The authors present data relating to multigenic screening in individuals with HBOC and negative for BRCA1 / 2 mutations. Despite the small number of individuals enrolled, this paper could be a good starting point for a broader screening. 

I suggest to indicate the CHEK2 variant c.1036C>T in the table S1. In fact, despite Varsome and literature data classify it as probable pathogenetic, it is reported as VUS by Clinvar. It would also be interesting in this case to make an analysis of the segregation of this variant within the IC's family.

Author Response

First of all I would like to appreciate the time and attention to our manuscript.

Point 1. I suggest to indicate the CKEK2 variant c.1036C>T in the table S1. In fact, despite Varsome and literature data classify it as probable pathogenic, it is reported as a VUS by ClinVar. It would also be interesting in this case to make an analysis of the segregation of this variant within the IC's family.

Response. As you point out, we agree to classify this CHEK2 variant as VUS. It was already add to table S1 (ID patient 41863), but with different nomenclature (c.1165C>T; p.Arg389Cys). We proceed to correct this inconsistency in TableS1.

Regarding segregation analysis in the IC's family, we have reviewed the family tree and it is currently an uninformative family. Among individuals with breast cancer, only IC is alive. Anyway, we will make a follow up of this family.

*Additional comment: Following your recommendations, in order to clarify data presentation, in the revised manuscript we have changed "Table 2. Germline PV identified" by "Figure 2. Distribution of variants detected by Next Generation Sequencing panel of 41 genes".

Round 2

Reviewer 1 Report

The authors have corrected errors and have added to the manuscript. Technically the manuscript has been improved.

Author Response

First of all I would like to appreciate your valuable review to our manuscript.

According to Editor Notes and your review, we have corrected typographical errors and modified some grammatical expressions that could be confusing. These changes are marked up in the revised file. In addition, we have uploaded a revised Graphical Abstract after correcting a typographical error.